# Sclerostin and Its Involvement in the Pathogenesis of Idiopathic Scoliosis

**DOI:** 10.3390/jcm10225286

**Published:** 2021-11-14

**Authors:** Elias S. Vasiliadis, Dimitrios Stergios Evangelopoulos, Angelos Kaspiris, Christos Vlachos, Spyros G. Pneumaticos

**Affiliations:** 13rd Department of Orthopaedics, School of Medicine, National and Kapodistrian University of Athens, KAT Hospital, 16541 Athens, Greece; ds.evangelopoulos@gmail.com (D.S.E.); christosorto@gmail.com (C.V.); spirospneumaticos@gmail.com (S.G.P.); 2Laboratory of Molecular Pharmacology, Division for Orthopaedic Research, School of Health Sciences, University of Patras, 26504 Rion, Greece; angkaspiris@hotmail.com

**Keywords:** idiopathic scoliosis, sclerostin, osteocytes, β-catenin, Wnt signaling pathway

## Abstract

Idiopathic scoliosis is a disorder of unknown etiology. Bone biopsies from idiopathic scoliosis patients revealed changes at cellular and molecular level. Osteocytic sclerostin is downregulated, and serum level of sclerostin is decreased. Osteocytes in idiopathic scoliosis appear to be less active with abnormal canaliculi network. Differentiation of osteoblasts to osteocytes is decelerated, while Wnt/β-catenin signaling pathway is overactivated and affects normal bone mineralization that leads to inferior mechanical properties of the bone, which becomes susceptible to asymmetrical forces and causes deformity of the spinal column. Targeting bone metabolism during growth by stimulating sclerostin secretion from osteocytes and restoring normal function of Wnt/β-catenin signaling pathway could, in theory, increase bone strength and prevent deterioration of the scoliotic deformity.

## 1. Introduction

Idiopathic scoliosis is a disorder of unknown etiology. Numerous factors have been involved in its pathogenesis either by initiating or by contributing to the progression of the deformity [1]. Sclerostin has been recognized as a negative regulator of bone formation through inhibition of canonical Wnt signaling pathway in osteoblast lineage cells [2], and its role was first recognized in the study of two rare high bone mass disorders, namely sclerosteosis [3] and van Buchem’s disease [4]. Systematic research into sclerostin’s biology and mechanisms of action revealed that it is involved in the pathogenesis of many skeletal disorders.

Low bone mass is a common finding among idiopathic scoliosis patients, and osteopenia is considered a prognostic factor for curve progression [5,6]. In addition, structural changes of bone micro-structure have been reported, [7] suggesting that there are changes at the cellular and molecular level. Abnormal differentiation of osteoblasts and a reduction in the population of osteocytes were found in bone tissues from idiopathic scoliosis patients [8]. Osteocytes in idiopathic scoliosis appear to be less active with abnormal canaliculi network, which affects normal bone mineralization [9]. Furthermore, sclerostin expression was found decreased, while β-catenin was overexpressed in bone lining cells derived from patients with idiopathic scoliosis [9]. The present review article summarizes the current knowledge on sclerostin structure, expression, and regulation and also discusses its role in the pathogenesis of idiopathic scoliosis.

## 2. Sclerostin Structure, Expression, and Regulation

Sclerostin is a 190-amino-acid secreted glycoprotein that is encoded by SOST, a gene identified in sclerosteosis and van Buchem’s disease patients [10]. SOST gene is localized in chromosome 17q12–q21, [11] and encodes a 27 kDa–213 amino-acid pro-peptide, with the first 23 amino acids as part of a signal sequence for secretion. Sclerostin, the SOST gene product, is rarely detected as the expected 27 kDa protein, as it is secreted predominantly in a dimeric form [12]. It has two glucosylation sites and contains a cystin-knot motif that covers residues 80–167 [13], indicating that sclerostin is closely related to the DAN protein family although its amino-acid similarity to the members of the DAN family is rather limited [14]. A similar sequence of the cystin-knot is present in TGFb superfamily. DAN family proteins antagonize BMPs signaling [15]. Although the role of sclerostin as a direct BMPs antagonist is not clear [2], there is evidence that it downregulates BMPs action by preventing BMPs binding to its receptors [16]. Sclerostin is also considered as an inhibitor of the canonical Wnt signaling pathway that plays a central role in the regulation of bone growth and remodeling through its binding to the Wnt LRP 5/6 co-receptors.

Sclerostin is considered as an osteocyte specific protein [16]. The SOST gene has been detected in numerous other organs, such as the lung, kidney, aorta, heart, liver, and bone [13,15,17]. By using SOST promoter LacZ reporter, SOST gene expression was documented in the epididymis, pyloric sphincter, carotid arteries, and parts of the cerebellum [18]. The presence of congenital hand disorders in patients with sclerosteosis [3] suggests that SOST is expressed in the developing embryo. LacZ reporter was detected in the distal limb bud ectoderm and gradually is restricted to the digits as gestational age progresses [18].

SOST gene expression has also been detected in articular cartilage [16], mineralized hypertrophic chondrocytes [19], cementocytes [19], and osteoclasts [20], suggesting that sclerostin is expressed by terminally differentiated cells within mineralized matrices [19].

Although sclerostin is found in numerous tissues, it is predominately produced by osteocytes and is associated with high bone mass disorders in humans [3,4]. SOST knockout mice also appear with an increased bone mass phenotype due to enhanced osteogenesis [21]. These findings assign sclerostin as a key negative regulator of bone formation through inhibition of canonical Wnt signaling pathway and antagonizing BMPs.

Activation of canonical Wnt signaling pathway is initiated by ligands, such as Wnt proteins, which bind to Frizzled receptors, and its co-receptors of low-density lipoprotein-related receptor (LRP) proteins, LRP5 and LRP6. An inhibition of GSK-3β follows inside the cytoplasm, leading to accumulation of β-catenin, which then translocates into the nucleus and induces gene transcription [22]. Through this mechanism, differentiation of mesenchymal stem cells is controlled in favor of osteoblastic differentiation. Osteblastic differentiation predominates, apoptosis of osteoblast precursor cells is suppressed, and chondrogenic, myogenic, and adipogenic differentiation is restrained, indicating that canonical Wnt signaling pathway is essential for MSCs differentiation to osteoblast lineage cells. Numerous LRP5 mutations were found to reduce sclerostin binding and result in high bone mass diseases, similar to sclerosteosis. These findings further support the role of sclerostin in regulating Wnt activity [23,24].

Furthermore, in osteoblast lineage cells, the activation of canonical Wnt signaling pathway enhances the secretion of osteoprotegerin (OPG), the decoy receptor for receptor activator of nuclear factor kappa-B ligand (RANKL), and suppresses osteoclast differentiation of osteoclast precursors [22]. SOST knock-out mice showed an anabolic effect with increased bone formation without elevation of bone resorption markers [21]. The increase in bone mass corresponds to improved microstructure with relative proteoglycan content, lower apatite crystal maturity, and lower matrix mineralization, which are related with increased resistance to maximum load, stiffness, and energy to failure [25].

Additionally, in osteocytes, sclerostin increases the expression of cathepsin K, tartrate-resistant acid phosphatase (TRAP), and carbonic anhydrase-2 in vitro, which are proteins involved in bone resorption and remodeling of extracellular matrix [26].

There are two regions that regulate SOST gene expression: the upstream promoter region and the downstream enhancer, evolutionary conserve region 5(ECR5). Interestingly, ECR5 is required for sclerostin production only in osteocytes and not in cells from other tissues [27]. These regulatory regions are sites where numerous transcription factors bind in response to cytokines and growth factors. In van Buchem’s disease, there is a homozygous absence of ECR5 region, which has binding sites for the myocyte enhancer factor (Mef2c) transcription factor [28], suggesting that the binding of Mef2c to the ECR5 region of the SOST gene is important for SOST gene expression. The upstream promoter region has binding sequences for osteoblast transcription factor runt-related transcription factor 2 (Runx2) [29] and also contains a methylation site. Demethylation of this region during osteoblast-osteocyte transition results in increased SOST expression [30].

Reports regarding the role of vitamin D in regulation of SOST gene expression are not consistent. Vitamin D was found either to induce SOST expression [31] or to decrease the transcriptional activity of the SOST promoter [32].

Current data demonstrated that sclerostin was also implicated in bone fracture healing process and in the pathophysiology of osteoarthritis and spinal deformities or disc degeneration via its angiogenetic activities [33]. Micro-CT analysis for vasculogenesis demonstrated that application of sclerostin antibody was associated with neo-angiogenesis improvement during the second post-fractural week. On the contrary, there was not any difference regarding vascularization at the sixth or ninth week. Contrariwise, no significant improvement was detected in total vessel volume or diameter during the eighth week after the injury [34]. Recently, it was displayed that the application of sclerostin antibodies [35] were involved in the cartilage preservation and chondrocyte metabolism inhibiting the expression of catabolic angiogenetic molecules, like Vascular Endothelial Growth Factor (VEGF) [17]. In-vitro studies demonstrated that sclerostin significantly induces Primary Human Umbilical Vein Endothelial Cells (HUVEC) cell proliferation, an effect that is exerted by VEGF [36]. Moreover, sclerostin angiogenic activity in vivo was evaluated with chorioallantoic membrane (CAM) assay, and it was comparable to the VEGF-induced proangiogenic effects [36]. Similarly, the expression of anti-angiogenic Pigment Epithelium-Derived Factor (PEDF), whose down-regulation was closely associated with the progression of skeletal angiogenic deformities and was spatially expressed in areas of endochondral ossification and bone remodeling, was strongly inhibited by the expression of sclerostin [37]. These activities of sclerostin were also associated with increased recruitments of osteoclasts and their circulating monocyte progenitors and with reduced expression of osteoblastic genes and bone mineralization [36]. Based on these findings, it was suggested that sclerostin not only induced angiogenesis and can be classified with other Wnt antagonists already known for their angiogenic properties but can be considered as a critical growth factor for angiogenesis-osteogenesis coupling effect [36].

Sclerostin is expressed by osteocytes in mineralized bone and by mineralizing chondrocytes in articular cartilage and the growth plate in most primary bone tumors, such as osteoma, osteoid osteoma, osteoblastoma, and osteosarcoma [38]. Furthermore, sclerostin is expressed by tumor cells in most high-grade osteosarcomas and pariosteal osteosarcomas. In human osteosarcoma Saos2 cell line, Runx2 was found to increase SOST gene expression by binding to the proximal promoter [39]. The precise role of sclerostin is not clear, and evidence regarding the role of Wnt-signaling pathway in osteosarcoma is contradictory. Wnt-signaling pathway was reported either to facilitate osteosarcoma [40] or to suppress cellular proliferation and metastases [41].

Serum sclerostin levels were found to be age dependent [38,39]. Older women had 46% higher serum sclerostin levels than younger women [38]. Conversely, both young and aged women had similar bone SOST mRNA levels, implying that other tissues may be responsible for increased sclerostin secretion with age [38].

Furthermore, sclerostin was expressed by tumor cells in most high-grade osteosarcomas and parosteal osteosarcomas. In human osteosarcoma Saos2 cell line, Runx2 was found to increase SOST gene expression by binding to the proximal promoter [39]. The precise role of sclerostin is not clear and evidence regarding the role of Wnt-signaling pathway in osteosarcoma is contradictory. Wnt-signaling pathway was reported either to facilitate osteosarcoma [40], or to suppress cellular proliferation and metastases [41].

Serum sclerostin levels were found to be age dependant [42,43]. Older women had 46% higher serum sclerostin levels than younger women [42]. Conversely, both young and aged women had similar bone SOST mRNA levels, implying that other tissues may be responsible for increased sclerostin secretion with age [42].

## 3. Osteocytes in Idiopathic Scoliosis

In bone tissues from patients with idiopathic scoliosis, the population of bone lining cells was found different than controls. Number of osteocytes was lower, osteoblasts were higher, and osteoclasts were found unaltered [9]. Osteocytes are key regulators of bone remodeling by transforming mechanical forces into structural changes of bone microenvironment through regulation of bone homeostasis. Osteocytes are considered as the main mechano-sensors of bone, and their impact of mechano-transduction is determined by numerous signaling networks, which are involved in skeletal growth and function [44].

Osteocytes are located in cavities in mineralized matrix called lacunae and form a well-organized network of small canals, the canaliculi. Canaliculi are occupied by osteocyte cellular extensions, the dendrites, which allow communication with other cells and are filled by canalicular fluid, which carries nutrients and biological factors and transmits mechanical signals [45]. Mechanical loading exerted on bone matrix is amplified through fluid flow inside lacunae and canaliculi [46] and stimulates osteocytes, which transduce the mechanical signal into biochemical reaction [47]. In idiopathic scoliosis, osteocytes’ canaliculi were found disorganized and less in number, with shortening of their dendrites’ length [9].

In response to mechanical loading, osteocytes produce numerous biological factors that regulate bone homeostasis and remodeling. Following mechanical loading, osteocytes secrete sclerostin, prostaglandins [48], and other inflammatory factors, such as TNF, leukemia inhibitory factor, and oncostatin M, which downregulate sclerostin expression [49].

Abnormal osteocyte function in idiopathic scoliosis results in impaired bone mineralization and inferior mechanical properties of the bone, which is susceptible to asymmetric forces that cause deformity of the spine [50].

## 4. Sclerostin and Idiopathic Scoliosis

Osteocytic sclerostin is downregulated by mechanical loading in wild-type mice, resulting in increase of new bone formation after canonical Wnt-signaling pathway activation, while in human SOST transgenic mice (DMP1-SOST), the anabolic response to loading was disrupted [51]. Suppression of sclerostin after mechanical loading is caused, among other mechanisms, by the increased periostin secretion, by periosteal osteoblasts [52], and the involvement of prostaglandin E2 and nitric oxide [47,52].

Osteocytes from bone biopsies of patients with idiopathic scoliosis had decreased SOST gene expression as well as lower sclerostin secretion. Osteocytes derived from CTNNB1 knocking down osteoblasts from patients with idiopathic scoliosis SOST gene expression and sclerostin secretion were found significantly elevated [9]. These findings provide evidence that sclerostin antagonizes Wnt/β-catenin signaling pathway.

In bone marrow mesenchymal stem cells derived from idiopathic scoliosis patients, miRNAs miR-17-5p, miR-106a-5p, miR-106b-5p, miR-16-5p, miR-93-5p, and miR-181b-5p were found up-regulated. Their role was identified as osteogenic differentiation and bone-formation suppressors. Additionally, up-regulation of miR-15a-5p was involved in regulation of cell apoptosis [53]. There are several studies in the literature that negatively corelate miRNAs with sclerostin. Higher serum levels of miRNA-21 were coupled with lower levels of sclerostin [54], while inhibition of miRNA-218-5p promoted SOST gene expression [55]. Serum level of SOST was negatively correlated with plasma miR-145, and serum level of SOST miR-145 knockdown in osteoblasts from idiopathic scoliosis patients improved osteocyte function possibly by maturation of osteocytes and dendrite formation and not by osteoblast differentiation [9]. Decreased levels of sclerostin may be explicated due to elevated levels of miRNAs in bone biopsies from idiopathic scoliosis patients.

Wnt signaling pathway was found overactivated in bone biopsies from scoliotic patients as active β-catenin was found significantly elevated [50]. Similar findings were reported in zebrafish scoliotic model, where β-catenin activity was related to the deformity of the spine through the enzyme tyrosine kinase 7 [56]. Although Wnt/β-catenin activation increases bone mass, overactivation of Wnt/β-catenin signaling pathway in idiopathic scoliosis halt osteoblasts from differentiation to osteocytes [57] and compromise matrix mineralization [58]. Additionally, Runx2, an early bone formation marker, was found decreased in idiopathic scoliosis bone tissues, meaning that bone formation was impaired due to Wnt/β-catenin signaling pathway overexpression [7]. Overactivation of Wnt/β-catenin signaling pathway may play a role in the progression of the deformity through abnormal function of muscles [59], the intervertebral disc [60], and the vertebral growth plate [61]. One possible explanation of asymmetrical muscle contraction between convex and concave side of a scoliotic curve is the role of cadmodulin and its relation to Wnt/β-catenin signaling pathway and sclerostin expression. Cadmodulin concentration in paraspinal muscles was found increased at the convex side and decreased at the concave side of patients with idiopathic scoliosis [62]. Calmodulin after binding with calcium activates myosin light chain and contributes in smooth muscle contraction [63]. Cadmodulin downregulation was found to affect calcitonin and consequently calcium blood levels and G proteins activation [64]. Calcitonin downregulation subsequently downregulates sclerostin expression [65] and activates Wnt/β-catenin signaling pathway in osteoblastic differentiation of bone marrow stem cells [66]. Similarly, G proteins induce Wnt/β-catenin signaling [51] and show higher expression in the vertebral bodies from the convex side of the scoliotic spine [67]. One possible explanation of sclerostin downregulation in patients with idiopathic scoliosis is through the cadmodulin/calcitonin interaction.

Asymmetric mechanical forces over a vulnerable spinal column during growth may cause deformity of the spine. The origin of asymmetric mechanical forces is unknown. There are numerous etiologic concepts in the literature that involve relative anterior spinal overgrowth [68]; rib cage asymmetry [69]; handedness [70]; central nervous system abnormalities [71,72]; deficits of the vestibular system, which may affect the tone of paraspinal muscles [73]; decreased leptin levels in the peripheral blood [74]; and the double neuro-osseous theory, which results in the loss of co-ordination between autonomic and somatic nervous systems of the spine and trunk [75]. A direct association between mechanical loading and sclerostin expression is documented in the literature. Enhanced loading of the ulna in rodents reduced osteocyte expression of sclerostin, and parts of cortical bone sustaining greater strain demonstrated a more profound reduction of sclerostin staining osteocytes [76]. Numerous genes associated with activation of Wnt-signaling pathway in wild-type mice were upregulated after loading, but in DMP1-SOST mice, this response was not evidenced [77]. Mechanical loading was associated with inhibition of bone loss in response to unloading in SOST knock-out mice [78]. Stroke patients, who experience mechanical unloading due to long-term immobilization, were found with increased levels of serum sclerostin [79]. Anti-sclerostin antibodies prevented unloading induced bone loss in a hindlimb-immobilization rat model [80]. Mechanical loading was found to suppress TGFβ signaling in osteocytes, which is required for SOST gene expression, resulting in sclerostin suppression [81]. Although sclerostin-independent changes in bone formation after loading has been reported [82], sclerostin downregulation is critical for the osteogenic response to mechanical loading by relieving inhibition of Wnt-signaling pathway. Although increased mechanical forces decrease sclerostin expression from osteocytes, there is no evidence for how asymmetrical loading may affect sclerostin in idiopathic scoliosis. Further studies are required to investigate this association.

Wnt/β-catenin signaling pathway overexpression and sclerostin downregulation by osteocytes could be progressive factors for the development of idiopathic scoliosis. Although sclerostin is a negative regulator of bone formation, it could play a role in Wnt/β-catenin signaling pathway downregulation. In theory, by restoring bone metabolism during growth at a molecular level through Wnt/β-catenin signaling pathway downregulation and stimulating sclerostin secretion from osteocytes at a cellular level, bone vulnerability could be alleviated, and the scoliotic deformity could be improved. Further studies are needed in order to test this hypothesis, and care should be taken to minimize the possible side effects of Wnt/β-catenin signaling pathway and sclerostin action at sites other than the spinal column.

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
