# Peer review of "Sclerostin and Its Involvement in the Pathogenesis of Idiopathic Scoliosis"

_jcm, 2021, doi:10.3390/jcm10225286_

Round 1

Reviewer 1 Report

This is an interesting paper reviewing the role of sclerostin in osteocyte cell metabolism. It is divided into three parts to facilitate the reader's understanding: the structure of sclerostin and its relationship to osteocyte cellular mechanisms; the alterations observed in osteocytes in idiopathic scoliosis; and thirdly, the possible role of sclerotin in the development and progression of scoliosis. Overall I find it an interesting and well-written paper. However, there are some statements in the text that deserve comment.

In the Introduction (page 1 line 40) they stated: "The present review article .... it discusses its role (sclerostin) in the pathogenesis of idiopathic scoliosis". However, this "discussion" seems to me to be rather poor.

"Overactivation of Wnt/β-catenin signaling pathway may play a role in the progression of the deformity through abnormal function of muscles". This is a very interesting point as idiopathic scoliosis has been related to some unclear form of subclinical myopathy. In this respect I would like to recall the interest in the effect of calmodulin on muscle contractility. Is there a relationship between sclerotin and calmodulin?

"Asymmetric mechanical forces over a vulnerable spinal column during growth may cause deformity of the spine". At this point there is no doubt, but what is the reason for a straight and symmetrical column to become asymmetrical?

"Wnt/β-catenin signaling pathway overexpression and sclerostin downregulation by osteocytes could be progressive factors for the development of idiopathic scoliosis". Are you suggesting that a minor asymmetry could be self-correcting if there were no overexpression of the Wnt/β-catenin signaling pathway?

We must assume that the cellular alteration of the osteocytes affects all bones of the skeleton. Why would only the vertebrae be manifestly affected?

"Furthermore, measurement of β-catenin and sclerostin could be possibly be used as biomarkers for the prognosis of curve progression over time". I am afraid that this claim is not based on any evidence and seems to me to be extremely daring. 

Author Response

This is an interesting paper reviewing the role of sclerostin in osteocyte cell metabolism. It is divided into three parts to facilitate the reader's understanding: the structure of sclerostin and its relationship to osteocyte cellular mechanisms; the alterations observed in osteocytes in idiopathic scoliosis; and thirdly, the possible role of sclerotin in the development and progression of scoliosis. Overall I find it an interesting and well-written paper. However, there are some statements in the text that deserve comment.

Point 1: In the Introduction (page 1 line 40) they stated: "The present review article .... it discusses its role (sclerostin) in the pathogenesis of idiopathic scoliosis". However, this "discussion" seems to me to be rather poor.

Response 1: All the suggestion by the reviewers were added in the discussion section as well as the relative references.

Point 2: "Overactivation of Wnt/β-catenin signaling pathway may play a role in the progression of the deformity through abnormal function of muscles". This is a very interesting point as idiopathic scoliosis has been related to some unclear form of subclinical myopathy. In this respect I would like to recall the interest in the effect of calmodulin on muscle contractility. Is there a relationship between sclerotin and calmodulin?

Response 2: Although there is no direct relation of sclerostin and calmodulin in patients with idiopathic scoliosis in the literature, asymmetric expression of calmodulin affects the expression of Wnt/β-catenin signaling pathway and consequently bone formation between the convex and concave side of a scoliotic spine. A relative paragraph was added in the discussion section.

Point 3: "Asymmetric mechanical forces over a vulnerable spinal column during growth may cause deformity of the spine". At this point there is no doubt, but what is the reason for a straight and symmetrical column to become asymmetrical?

Response 3: The origin of asymmetric mechanical forces is unknown. A summary of most theories that explain why a straight and symmetrical column becomes asymmetrical are briefly added in the discussion section.

Point 4: "Wnt/β-catenin signaling pathway overexpression and sclerostin downregulation by osteocytes could be progressive factors for the development of idiopathic scoliosis". Are you suggesting that a minor asymmetry could be self-correcting if there were no overexpression of the Wnt/β-catenin signaling pathway?

Response 4: In theory, yes. But further studies are required in order to support this hypothesis.

Point 5: We must assume that the cellular alteration of the osteocytes affects all bones of the skeleton. Why would only the vertebrae be manifestly affected?

Response 5: This is a very interesting point. No-one can exclude possible side effects of an osteocyte – targeting therapy of idiopathic scoliosis. Anti-sclerostin antibody, which has been approved for the treatment of osteoporosis has similar side effects, because it does not affect only bone metabolism. A relative sentence was added in the discussion section.

Point 6: "Furthermore, measurement of β-catenin and sclerostin could be possibly be used as biomarkers for the prognosis of curve progression over time". I am afraid that this claim is not based on any evidence and seems to me to be extremely daring. 

Response 6: The sentence has been removed.

Reviewer 2 Report

Vasiliadis et al. survey the existing literature that demonstrates possible role of sclerostin in idiopathic scoliosis.  This is particular interest as patients with idiopathic scoliosis often have osteopenia, which increases the fracture risk.  Currently available drug such as romosozumab may be a way to treat patients with idopathic scoliosis. 

The review provide sufficient background in terms of sclerostin structure, function, and Wnt signaling.  Unfortunately the review does not appear to go in-depth in terms of the role of sclerostin in idiopathic scoliosis.  For example in the introduction, authors demonstrate that idiopathic scoliosis patients often have osteopenia.  However, in section 4, authors also demonstrate that bone biopsies from those patients show decreased SOST gene expression, which implies there should be increased bone.  There is also insufficient discussion on how the loss/gain of function of Sost gene affect spinal curvature in mice.  Even if there is a lack of literature, authors can potentially deduce its role by assess the downstream targets of Sost gene.  As such despite authors attempt to survey the literature, the extent of knowledge that the readers will gain regarding the role of sclerostin in idiopathic scoliosis seems very limited. 

Also this review lacks discussion in terms of what is the next research frontier in terms of identifying the role of sclerostin in idopathic scoliosis.  Will it be more clinically based research to identify the effects of romosozumab?  Or are there any underlying mechanisms that needs to be probed further?  These are some of the discussion that will inform the readers in terms of existing knowledge gaps and how we can bridge those. 

Author Response

Vasiliadis et al. survey the existing literature that demonstrates possible role of sclerostin in idiopathic scoliosis.  This is particular interest as patients with idiopathic scoliosis often have osteopenia, which increases the fracture risk.  Currently available drug such as romosozumab may be a way to treat patients with idopathic scoliosis. 

Point 1: The review provide sufficient background in terms of sclerostin structure, function, and Wnt signaling.  Unfortunately the review does not appear to go in-depth in terms of the role of sclerostin in idiopathic scoliosis.  For example in the introduction, authors demonstrate that idiopathic scoliosis patients often have osteopenia.  However, in section 4, authors also demonstrate that bone biopsies from those patients show decreased SOST gene expression, which implies there should be increased bone.  There is also insufficient discussion on how the loss/gain of function of Sost gene affect spinal curvature in mice.  Even if there is a lack of literature, authors can potentially deduce its role by assess the downstream targets of Sost gene.  As such despite authors attempt to survey the literature, the extent of knowledge that the readers will gain regarding the role of sclerostin in idiopathic scoliosis seems very limited. 

Response 1: In idiopathic scoliosis it seems that there is an overactivation of Wnt/β-catenin signaling pathway which explains the elevated levels of β-catenin and the decreased levels of sclerostin. This overexpression has opposite effects in bone metabolism and instead of promoting bone formation it inhibits both differentiation of osteoblasts and bone mineralization. All the above clearly explain the osteopenia in patients with idiopathic scoliosis and are analyzed in the discussion section.

Point 2: Also this review lacks discussion in terms of what is the next research frontier in terms of identifying the role of sclerostin in idopathic scoliosis.  Will it be more clinically based research to identify the effects of romosozumab?  Or are there any underlying mechanisms that needs to be probed further?  These are some of the discussion that will inform the readers in terms of existing knowledge gaps and how we can bridge those. 

Response 2: A sentence was added at the end of the relevant paragraph in the discussion section to highlight the need for further research about the role of bone metabolism restoration during growth at a molecular level through Wnt/β-catenin signaling pathway downregulation and sclerostin secretion stimulation from osteocytes at a cellular level. Romosozumab and other anti-sclerostin antibodies have opposite action by blocking sclerostin and therefore their potential role in studies for idiopathic scoliosis treatment are inappropriate.

Reviewer 3 Report

Thank you for the opportunity to review this work, which attempts to make a contribution to a complex and still unfortunately unclear area such as the etiology of idiopathic scoliosis. The manuscript is well written and the topic interesting. However I have some concerns that deserve some clarification.

At line 39 you said that sclerostin was found over-expressed in osteocytes from patients with idiopathic scoliosis (“sclerostin expression was found increased”). However, in the abstract and throughout the text you discussed the potential role of sclerostin down-regulation in scoliosis. Also on line 172 you reported: “Osteocytes from bone biopsies of patients with idiopathic scoliosis had decreased SOST gene expression as well as lower sclerostin secretion”. Can you please clarify this point?

At line 192 you said: “Although sclerostin is a negative regulator of bone formation, it could play a role in Wnt/β-catenin signaling pathway downregulation”. At line 173 you said: “Osteocytes derived from CTNNB1 knocking down osteoblasts from patients with idiopathic scoliosis SOST gene expression and sclerostin secretion were found significantly elevated”. From these sentences I understand that the lack of beta-catenin may correlate with the up-regulation of sclerostin. This would be in line with evidence that finds sclerostin and the WNT pathway regulated in an opposite direction. However, the sentence is difficult to understand and I would ask you to rephrase it to make it clearer. Also, I do not understand the sentence at line 175: “Furthermore, in idiopathic scoliosis patients serum level of SOST was negatively correlated with plasma miR-145 and serum level of SOST”. Anyway, the discussion of the relevance of miR-145 is limited to the sentence I quoted above and a sentence in the conclusions. I would ask you to remove the sentence in the conclusions and further discuss miR-145 in the body of the text.

At line 199 you said: “measurement of β-catenin and sclerostin could be possibly be used as biomarkers for the prognosis of curve progression over time”. This point is very relevant in terms of potential clinical applications. In clinical practice, how could the expression of these markers be assessed? What could be the alternatives to bone biopsy for their measurement?

Finally, a curiosity of mine. Do you have knowledge of any studies that have evaluated a possible role of sclerostin in the oncogenesis of bone sarcomas? If so, a brief mention in the text might be helpful for a more global view of this gene.

Author Response

Thank you for the opportunity to review this work, which attempts to make a contribution to a complex and still unfortunately unclear area such as the etiology of idiopathic scoliosis. The manuscript is well written and the topic interesting. However I have some concerns that deserve some clarification.

Point 1: At line 39 you said that sclerostin was found over-expressed in osteocytes from patients with idiopathic scoliosis (“sclerostin expression was found increased”). However, in the abstract and throughout the text you discussed the potential role of sclerostin down-regulation in scoliosis. Also on line 172 you reported: “Osteocytes from bone biopsies of patients with idiopathic scoliosis had decreased SOST gene expression as well as lower sclerostin secretion”. Can you please clarify this point?

Response 1: The statement at line 39 is wrong and is corrected to “Sclerostin expression was found decreased”

Point 2: At line 192 you said: “Although sclerostin is a negative regulator of bone formation, it could play a role in Wnt/β-catenin signaling pathway downregulation”. At line 173 you said: “Osteocytes derived from CTNNB1 knocking down osteoblasts from patients with idiopathic scoliosis SOST gene expression and sclerostin secretion were found significantly elevated”. From these sentences I understand that the lack of beta-catenin may correlate with the up-regulation of sclerostin. This would be in line with evidence that finds sclerostin and the WNT pathway regulated in an opposite direction. However, the sentence is difficult to understand and I would ask you to rephrase it to make it clearer. Also, I do not understand the sentence at line 175: “Furthermore, in idiopathic scoliosis patients serum level of SOST was negatively correlated with plasma miR-145 and serum level of SOST”. Anyway, the discussion of the relevance of miR-145 is limited to the sentence I quoted above and a sentence in the conclusions. I would ask you to remove the sentence in the conclusions and further discuss miR-145 in the body of the text.

Response 2: A sentence added in line 200 which clarifies the antagonist role of sclerostin in Wnt/β-catenin signaling pathway. The sentence regarding miR-145 has been removed from the conclusions and further discussion was added in the body of the text.

Point 3: At line 199 you said: “measurement of β-catenin and sclerostin could be possibly be used as biomarkers for the prognosis of curve progression over time”. This point is very relevant in terms of potential clinical applications. In clinical practice, how could the expression of these markers be assessed? What could be the alternatives to bone biopsy for their measurement?

Response 3: The sentence has been removed.

Point 4: Finally, a curiosity of mine. Do you have knowledge of any studies that have evaluated a possible role of sclerostin in the oncogenesis of bone sarcomas? If so, a brief mention in the text might be helpful for a more global view of this gene.

Response 4: A paragraph concerning the role of sclerostin in bone sarcomas was added in the section “Sclerostin structure, expression and regulation”, lines 155-164.

Round 2

Reviewer 2 Report

The revised manuscript still lacks sufficient information regarding the role of sclerostin and idiopathic scoliosis.  Authors have included additional mechanisms that are altered in idiopathic scoliosis, such as miRs, calmodulin, calcitonin, and mechanical forces.  But authors fail to mention how these factors lead to changes in sclerostin.  Authors superficially mention there is an interaction with wnt/b-catenin signaling, but the focus on section 4 is sclerostin and idiopathic scoliosis.  In that sense, readers are more interested in how these factors lead to changes in sclerostin level, not necessarily wnt/b-catenin signaling as a whole. 

Also, the author's response re overexpression of wnt/b-catenin signaling leads to some doubts whether they have sufficient knowledge to write this review.  Multiple studies (Dao et al. 2010; Yan et al. 2009; Kugimiya et al. 2007 to name a few) have demonstrated that overactivation of wnt/b-catenin signaling increases bone mass and osteoblast proliferation.  Or perhaps this suggests that idiopathic scoliosis and Wnt/b-catenin signaling is not really related. 

Author Response

Reviewer 2

Point 1: The revised manuscript still lacks sufficient information regarding the role of sclerostin and idiopathic scoliosis.  Authors have included additional mechanisms that are altered in idiopathic scoliosis, such as miRs, calmodulin, calcitonin, and mechanical forces.  But authors fail to mention how these factors lead to changes in sclerostin.  Authors superficially mention there is an interaction with wnt/b-catenin signaling, but the focus on section 4 is sclerostin and idiopathic scoliosis.  In that sense, readers are more interested in how these factors lead to changes in sclerostin level, not necessarily wnt/b-catenin signaling as a whole. 

Response 1: All the additional mechanisms were added according to the suggestions of the reviewers. In the revised section 4 ‘the role of sclerostin in idiopathic scoliosis’, a thorough analysis of how the additional mechanisms (miRNA, calmodulin, calcitonin and mechanical forces) lead to sclerostin changes in patients with idiopathic scoliosis with the relative literature is provided.

Additionally, most of sclerostin's actions in the musculoskeletal system are strongly associated with Wnt/β-catening signaling pathway. Wnt/β-catenin signaling pathway is an integral part of sclerostin's role and must be included in the study.

Point 2: Also, the author's response re overexpression of wnt/b-catenin signaling leads to some doubts whether they have sufficient knowledge to write this review.  Multiple studies (Dao et al. 2010; Yan et al. 2009; Kugimiya et al. 2007 to name a few) have demonstrated that overactivation of wnt/b-catenin signaling increases bone mass and osteoblast proliferation.  Or perhaps this suggests that idiopathic scoliosis and Wnt/b-catenin signaling is not really related. 

Response 2: It is a common knowledge and not a matter of expertise that overactivation of wnt/b-catenin signaling increases bone mass and osteoblast proliferation. There are numerous studies in the literature supporting that. The authors do not claim the opposite. Instead they highlight the paradox which is evident in idiopathic scoliosis, where overactivation of Wnt/β-catenin signaling pathway has opposite effects in bone metabolism and instead of promoting bone formation it inhibits both differentiation of osteoblasts and bone mineralization. All the above clearly explain the osteopenia in patients with idiopathic scoliosis and are analyzed in the discussion section, where the relative literature is provided.